# Cholinergic deficits selectively boost cortical intratelencephalic control of striatum in male Huntington's disease model mice

Tristano Pancani[1], Michelle Day[1], Tatiana Tkatch[1], David L. Wokosin[1], Patricia González-Rodríguez [1,2], Jyothisri Kondapalli[1], Zhong Xie [1], Yu Chen[1], Vahri Beaumont[3] & D. James Surmeier [1] ✉

Huntington's disease (HD) is a progressive, neurodegenerative disease caused by a CAG triplet expansion in huntingtin. Although corticostriatal dysfunction has long been implicated in HD, the determinants and pathway specificity of this pathophysiology are not fully understood. Here, using a male zQ175[+/−] knock-in mouse model of HD we carry out optogenetic interrogation of intratelencephalic and pyramidal tract synapses with principal striatal spiny projection neurons (SPNs). These studies reveal that the connectivity of intratelencephalic, but not pyramidal tract, neurons with direct and indirect pathway SPNs increased in early symptomatic zQ175[+/−] HD mice. This enhancement was attributable to reduced pre-synaptic inhibitory control of intratelencephalic terminals by striatal cholinergic interneurons. Lowering mutant huntingtin selectively in striatal cholinergic interneurons with a virally-delivered zinc finger repressor protein normalized striatal acetylcholine release and intratelencephalic functional connectivity, revealing a node in the network underlying corticostriatal pathophysiology in a HD mouse model.

Huntington's disease (HD) is a multifaceted, neurodegenerative disease caused by a CAG expansion in *huntingtin*. After an extended prodromal period, patients manifest progressive worsening of motor control and cognitive abilities[1]. Although mutant huntingtin (mHtt) is widely distributed, HD neuropathology is particularly prominent in the striatum[2], a brain region belonging to the basal ganglia circuitry that regulates goal-directed actions, habit, and cognition[3–5].

The principal neurons of the striatum are GABAergic spiny projection neurons (SPNs). There are two major classes of SPN, so-called direct pathway SPNs (dSPNs) and indirect pathway SPNs (iSPNs); these two types of SPN anchor basal ganglia networks that differentially control movement[6,7]. The activity of both iSPNs and dSPNs is controlled by axospinous, excitatory glutamatergic synapses formed by projection neurons from the cerebral cortex. In HD models, a great deal of attention has been focused on the role of corticostriatal

dysfunction in the evolution of striatal pathophysiology. Indeed, there is compelling evident that mHtt impairs corticostriatal transport, release, and signaling of brain derived neurotrophic factor (BDNF), particularly that impinging upon iSPN involved in the suppression of unwanted movement[8,9]. Consistent with this inference, electrophysiological, and anatomical studies have revealed an apparent attenuation in the corticostriatal connectivity in HD models[10–16]. However, there are several reasons to think that the impact of mHtt on corticostriatal circuitry is not this circumscribed.

One of the shortcomings of previous studies has been the reliance upon electrical stimulation to drive corticostriatal activity. This approach does not allow an assessment of whether the striatal innervation arising from intratelencephalic (IT) and pyramidal tract (PT) cortical neurons is affected in the same way by mHtt[17]. The projection systems anchored by these neurons undoubtedly play different roles

[1]Department of Neuroscience, Feinberg School of Medicine, Northwestern University, Chicago, IL 60613, USA. [2]Department of Medical Physiology and Biophysics Instituto de Biomedicina de Sevilla (IBiS), 41013 Sevilla, Spain. [3]CHDI Management/CHDI Foundation, Suite 700, 6080 Center Drive, Los Angeles, CA 90045, USA. ✉e-mail: j-surmeier@northwestern.edu

in motor control[18]. PT signaling to the striatum is thought to convey information about ongoing motor execution, whereas IT signaling is related to motor planning, context, and internal state[19,20]. Distinguishing between them could prove to be critical to understanding striatal HD pathophysiology, particularly in light two additional observations. The release of glutamate at corticostriatal axospinous synapses is controlled by presynaptic G-protein coupled receptors (GPCRs) and by ionotropic nicotinic ACh receptors (nAChRs)[21–24]. One of the most prominent of the GPCRs at these terminals is the inhibitory $G_{i/o}$-coupled muscarinic acetylcholine receptor (mAChR) that—like the nAChR—is activated by acetylcholine (ACh) release by giant, aspiny striatal cholinergic interneurons (ChIs)[21,24,25]. Interestingly, activation of nAChRs facilitate glutamate release selectively at PT synapses[22], raising the possibility that the prominent mAChR-mediated suppression of corticostriatal glutamate release seen with electrical stimulation of axons occurs only at IT synapses. Added motivation for pursuing this connection comes from the observation that, although they are not lost in HD, ChIs are functionally compromised[26–29], suggesting that striatal pathophysiology in HD stems in part from an imbalance in IT and PT control of SPNs.

To rigorously pursue this possibility, optogenetic approaches were used in the zQ175[+/−] mouse model of HD[30] to selectively interrogate the functional connectivity of IT and PT corticostriatal terminals with identified iSPNs and dSPNs. Surprisingly, these experiments revealed that functional synaptic connectivity of M1, M2 and cingulate cortex IT neurons with both iSPN and dSPNs was elevated in the dorsal striatum of zQ175[+/−] mice. This change was traced to a deficit in activity-

dependent ACh release by ChIs that attenuated mAChR depression of IT terminal glutamate release. Indeed, reduction in mHtt selectively in ChIs with viral delivery of zinc finger proteins (ZFPs)[31] normalized corticostriatal synaptic transmission, but lowering mHtt in M1 cortex or in iSPNs did not.

## Results

### The connectivity of IT but not PT neurons with SPNs was elevated in zQ175[+/−] mice

A major shortcoming of previous studies focused on corticostriatal connectivity in mouse models of HD has been the reliance upon electrical stimulation in ex vivo brain slices[10,12–14,26,32–36]. In the slice preparation, corticostriatal axons are invariably cut, limiting evoked responses with an electrical stimulus, and necessitating placement of the electrode close to or within the striatum; this placement allows current spread to the striatum itself, activating axons and neurons within this region. Not only does this make interpretation of changes in evoked responses problematic, it does not allow interrogation of specific corticostriatal networks. To overcome these limitations, optogenetic approaches were used in heterozygous zQ175[+/−] mice to assess changes in the functional connectivity of IT and PT neurons with iSPNs and dSPNs in the dorsolateral striatum (DLS).

To measure IT synaptic transmission, an AAV vector carrying a synapsin-driven Chronos-GFP construct (0.2 μl; AAV9.hSyn.Chronos-GFP; Figs. 1a, S1a) was injected into the M1 motor cortical region of 8–9 month-old zQ175[+/−] and zQ175[−/−] littermate control mice that expressed either tdTomato under control of the D1 receptor promoter, or eGFP

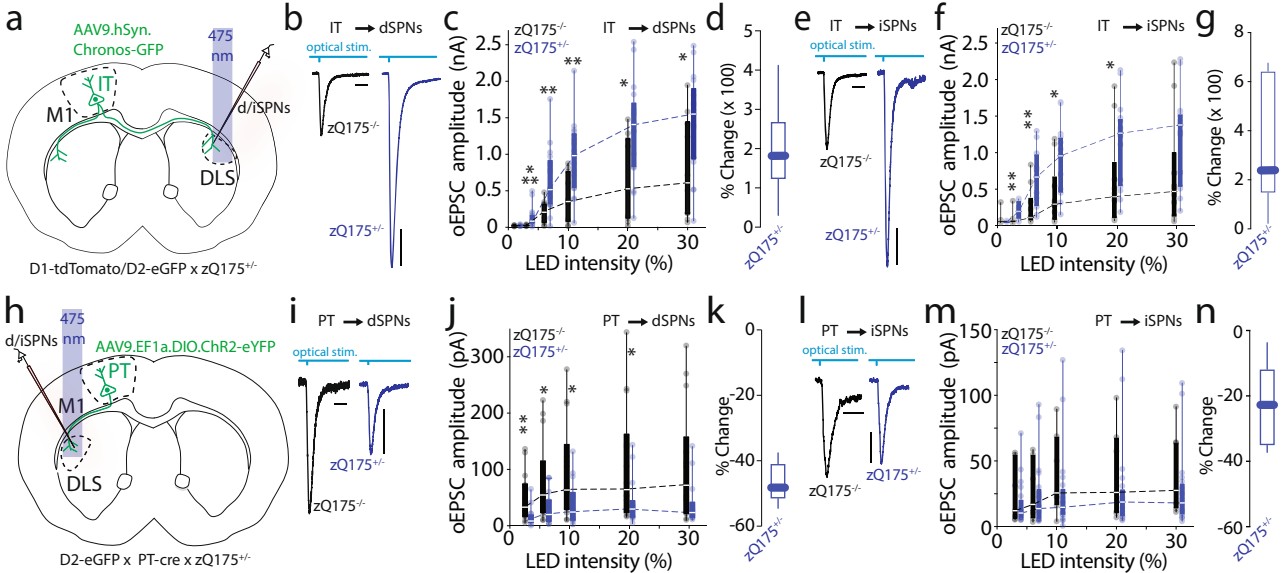

**Fig. 1 | Increased connectivity at intratelencephalic (IT)-SPNs synapses with no change or loss of PT connectivity in male zQ175[+/−] mice. a** Schematics of AAV9.hSyn.Chronos-GFP injected in primary motor cortex (M1) in a double transgenic mouse obtained crossing D1-tdTomato or D2-eGFP with zQ175[+/−] mice. Whole-cell recording from spiny projection neurons (SPNs) in dorsolateral striatum (DLS) in the contralateral hemisphere. **b, e** Representative traces of optically evoked EPSCs (oEPSC) recorded from direct pathway SPNs (dSPNs) (**b**) and indirect pathway SPNs (iSPNs) (**e**) (Scale bars: 10 ms and 200 pA). **c, f** Graphs showing increased oEPSCs amplitude when plotted against LED intensity in zQ175[+/−] mice compared to zQ175[−/−] littermate controls. Input/output curves (I/O) in dSPNs (**c**; zQ175[−/−] controls (N = 5 mice; n = 10 neurons) vs. zQ175[+/−] (N = 4; n = 17); p-value (LED Intensity, % of max): n.s. (1%), ***0.0004 (3%), **0.0047 (6%), *0.0470 (10%), *0.0110 (20%), *0.0350 (30%)), and iSPNs (**e**; zQ175[−/−] (N = 5; n = 11) vs. zQ175[+/−] (N = 4; n = 15); p-value (LED Intensity, % of max): n.s. (1%), **0.0054 (3%), **0.0036 (6%), *0.0163 (10%), *0.0475 (20%), n.s. 0.0623 (30%)). **d, g** Boxplots showing % change of oEPSC

amplitudes in zQ175[+/−] compared to zQ175[−/−] recorded at 30% LED intensity from (**c**) and (**f**) (**d**: n = 17; **g**: 15 cells). **h** Schematic representation of the AAV9.EF1a.-DIO.ChR2-GFP injection in a triple transgenic mouse obtained crossing D2-eGFP with PT-cre, and zQ175[+/−] mice (see also Fig. S1r). Whole-cell recording performed in ipsilateral DLS. **i, l** Representative traces of oEPSCs recorded from SPNs (Scale bars: 10 ms and 10 pA for dSPNs; 10 ms and 20 pA iSPNs). **j, m** Graphs summarizing oEPSCs peak amplitude data plotted against LED intensity from I/O curves recorded in dSPNs (zQ175[−/−] controls (N = 3; n = 14), zQ175[+/−] (N = 7; n = 14), p-value (Stim. Intensity, % of max): **0.0580 (3%), *0.0186 (6%), *0.0182 (10%), *0.0241 (20%), 0.0556 (30%)), and iSPNs (zQ175[−/−] (N = 3; n = 7), zQ175[+/−] (N = 7; n = 20), p-value (Stim. Intensity, % of max): n.s. (1%), n.s. (3%), n.s. (6%), n.s. (10%), n.s. (20%), n.s. (30%)). **k, n** Boxplots showing % change of oEPSC amplitudes in zQ175[+/−] compared to zQ175[−/−] recorded at 30% LED intensity from (**j**) and (**m**) (**k**: n = 14; **n**: 20 cells). **c, d, f, g, j, l–n** Boxplots represent median and interquartile range, whiskers min/max value. All data tested using a Mann–Whitney non-parametric two-sided tests.

under the D2 receptor promoter (allowing identification of dSPNs and iSPNs, respectively). To measure IT connectivity, whole-cell recordings with a Cs$^+$-based internal solution were performed 20–30 days later from visually identified dSPNs and iSPNs in DLS contralateral to the AAV injection site. Unexpectedly, optically evoked excitatory post-synaptic currents (oEPSCs) from both zQ175$^{+/-}$ dSPNs and iSPNs were significantly larger than those in age-matched zQ175$^{-/-}$ littermate controls (Fig. 1b–g). A similar effect was seen when monitoring IT connectivity in SPNs from ipsilateral DLS using the Tlx3(PL56)-Cre line (a mouse line expressing Cre in IT neurons) crossed into the zQ175$^{+/-}$ line (Fig. S1b–e). The difference in IT connectivity was not evident in experiments with younger (2–3-month-old) mice, suggesting that the alteration in signaling developed in parallel with motor symptoms in this HD model[30,37] (Fig. S1f–i).

To determine whether this shift in connectivity was restricted to M1 IT projections, the functional connectivity of M2 motor cortex with DLS iSPNs was studied. Again, IT connectivity with iSPNs was elevated in slices from zQ175$^{+/-}$ mice (Fig. S1j–m). To determine if the change was restricted to the DLS, the IT projection from the cingulate cortex (Cng.Ctx.) onto iSPNs in the dorsomedial striatum (DMS) was examined. Once more, IT connectivity with iSPNs was more robust in tissue from zQ175$^{+/-}$ mice (Fig. S1n–q).

In contrast to IT neurons, PT neurons project exclusively to the ipsilateral striatum, where they synapse on both dSPNs and iSPNs[17]. To assess the functional connectivity of PT neurons with SPNs, the Sim1-(KJ18)-Cre transgenic mouse line in which Cre recombinase is expressed in layer 5 PT neurons (PT-cre)[38] was used. Drd2-eGFP x zQ175$^{+/-}$ mice were crossed with the Sim1-KJ18 mice and then an AAV carrying a Cre-dependent, synapsin ChR2-eYFP expression construct was injected into the M1 motor cortex (Figs. 1h, S1r; 0.4 µl, AAV9.EF1a.DIO.ChR2-eYFP). Thirty days post injection, oEPSCs were measured in DLS's dSPNs (eGFP negative) and neighboring iSPNs (eGFP positive) ipsilateral to the injection site. In contrast to the IT connectivity, the functional connectivity of PT neurons with dSPNs was significantly decreased (Fig. 1i–k) in slices from 8–9-month-old zQ175$^{+/-}$ mice, but connectivity with iSPNs was not significantly changed (Fig. 1l–n).

### The change in IT functional connectivity was attributable to a presynaptic mechanism

As a first step toward assessing the mechanism responsible for the elevation in IT responsiveness in zQ175$^{+/-}$ SPNs, an estimate of the relative density of α-amino-3-hydroxy-5-methyl-4-isoxazolepropionic acid receptors (AMPARs) and N-methyl-d-aspartate receptors (NMDARs) at IT synapses was generated. Postsynaptic long-term potentiation (LTP) of glutamatergic synapses is usually accompanied by an increase in the AMPAR to NMDAR ratio, as AMPAR are trafficked into the synapse[39]. Optogenetic methods, as described above, were used to selectively activate IT synapses on Cs$^+$-filled iSPNs or dSPNs held at either −70 mV to estimate AMPAR currents or at +40 mV to estimate NMDAR currents (Fig. S2a). The ratio of these two currents was indistinguishable in SPNs from zQ175$^{+/-}$ and zQ175$^{-/-}$ (Fig. S2b). Although it cannot be completely excluded[40], these observations do not support the proposition that the change in IT connectivity was attributable to postsynaptic, long-term potentiation.

If postsynaptic potentiation is set aside as an explanation for the change in IT responses, then there are two obvious alternatives: 1) the release probability at IT terminals increased or 2) the number of IT synapses increased. To explore the first possibility optogenetic strategies were employed using Chronos rather than ChR2 because of its ability to faithfully follow higher stimulation frequencies[41]. To determine if IT terminal glutamate release probability was altered in zQ175$^{+/-}$ SPNs, contralateral M1 IT axons were optogenetically stimulated with a paired pulse protocol (100 ms inter-stimulus interval) while recording in whole-cell voltage-clamp mode with a Cs$^+$-based internal solution to maximize space clamp. The relative amplitude of the EPSC evoked in

SPNs by the second of two optical stimuli was significantly smaller in zQ175$^{+/-}$ SPNs (Fig. 2a, b)—consistent with an elevation in the release probability of IT synapses.

To address the second possibility, subcellular channel-rhodopsin-assisted circuit mapping (sCRACM) was used to estimate the number of functional IT synapses on the dendrites of SPNs. In ex vivo brain slices from mice in which M1 neurons had been induced to express Chronos (as described in Fig. 1a), a blue laser beam (spot laser) was used to stimulate IT terminals synapsing on the spines of SPN dendrites in the contralateral striatum (Fig. 2c–e). These experiments were performed in the presence of tetrodotoxin (TTX) and 4-aminopyridine (4-AP) to block propagated activity. In addition, only distal dendrites were probed to avoid stimulating synapses above or below the plane of focus with the blue laser. Optically evoked excitatory postsynaptic currents (oEPSCs) were monitored with the somatic electrode (Fig. 2f)[6]. In the DLS from zQ175$^{-/-}$ controls mice, 5–20% of dSPNs and iSPNs spines had a detectable response to optogenetic activation of M1 IT terminals (Fig. 2g). In DLS SPNs from zQ175$^{+/-}$ mice, the proportion of spines having detectable responses was significantly greater (Fig. 2g). The oEPSC amplitude distributions in zQ175$^{+/-}$ were shifted towards larger amplitudes (Fig. 2h). Given the paired pulse data, these results are consistent with the proposition that the increased number of detectable IT synapses on SPNs dendrites reflects an increased probability of multi-vesicular release of glutamate and lack of post-synaptic receptor saturation at corticostriatal terminals[21], rather than there being more IT synapses.

### Enhanced IT functional connectivity was attributable to diminished acetylcholine release

Several types of G$_{i/o}$-coupled GPCRs inhibit glutamate release at corticostriatal synapses. Among the best described of these are M2-class mAChRs[21,25,42,43]. Because studies of human brains and preclinical rodent models suggest that there is a striatal cholinergic deficit in HD[27–29], our working hypothesis was that the enhanced functional connectivity of IT synapses was attributable to loss of pre-synaptic mAChR modulation. To directly assess whether ACh release was altered in the zQ175$^{+/-}$ striatum, a genetically-encoded optical sensor for ACh (AChSnFr[44]) was packaged in an AAV and stereotaxically delivered in DLS (Fig. 3a, b). A month later, ex vivo slices were prepared for imaging. In agreement with previous work, ACh release—evoked by a burst of electrical stimulations mimicking the naturally occurring low-frequency ChIs spiking activity (2 Hz)—was significantly lower in striata from zQ175$^{+/-}$ mice than in striata from interleaved zQ175$^{-/-}$ littermates (Fig. 3c, d). This deficit was developmentally regulated, as it was not evident in younger zQ175$^{+/-}$ mice (Fig. S9), where IT functional connectivity was normal (Fig. S1f–i). The reduction in evoked striatal ACh release was not peculiar to the zQ175 model, as it also was evident in the striata taken from another genetic model of HD—R6/2 mice—at an age when motor deficits were manifest (Fig. S3)[27–29].

Although M2-class mAChRs are known to inhibit glutamate release from corticostriatal terminals, it is not known whether this modulation is specific to IT or PT terminals. To address this question, PT projections to the striatum were induced to express ChR2 by stereotaxic injection of AAV9.EF1a.DIO.ChR2-eYFP into M1 cortex of Drd2-eGFP mice crossed in the PT-Cre line (Fig. 4a). A month later, the effect of the mAChR agonist Oxo-M (10 µM) on oEPSCs in ipsilateral SPNs in ex vivo slices was assessed in whole-cell voltage clamp recordings (Fig. 4b, c). The light intensity was adjusted to obtain oEPSCs that had peak amplitudes of about 100 pA to minimize clamping artifacts. Oxo-M had no effect on oEPSCs evoked by PT stimulation in either dSPNs or iSPNs (Figs. 4b, c, S4a), in agreement with recent work[22].

Next, the effects of mAChRs GPCRs on IT terminals was studied. To this end, AAV9.Syn.Chronos-GFP (0.2–0.3 µl) was injected into the M1 cortex of Drd1-tdTomato mice (Figs. 4d, S4b). A month later, ex vivo slices were prepared and whole-cell voltage clamp recordings

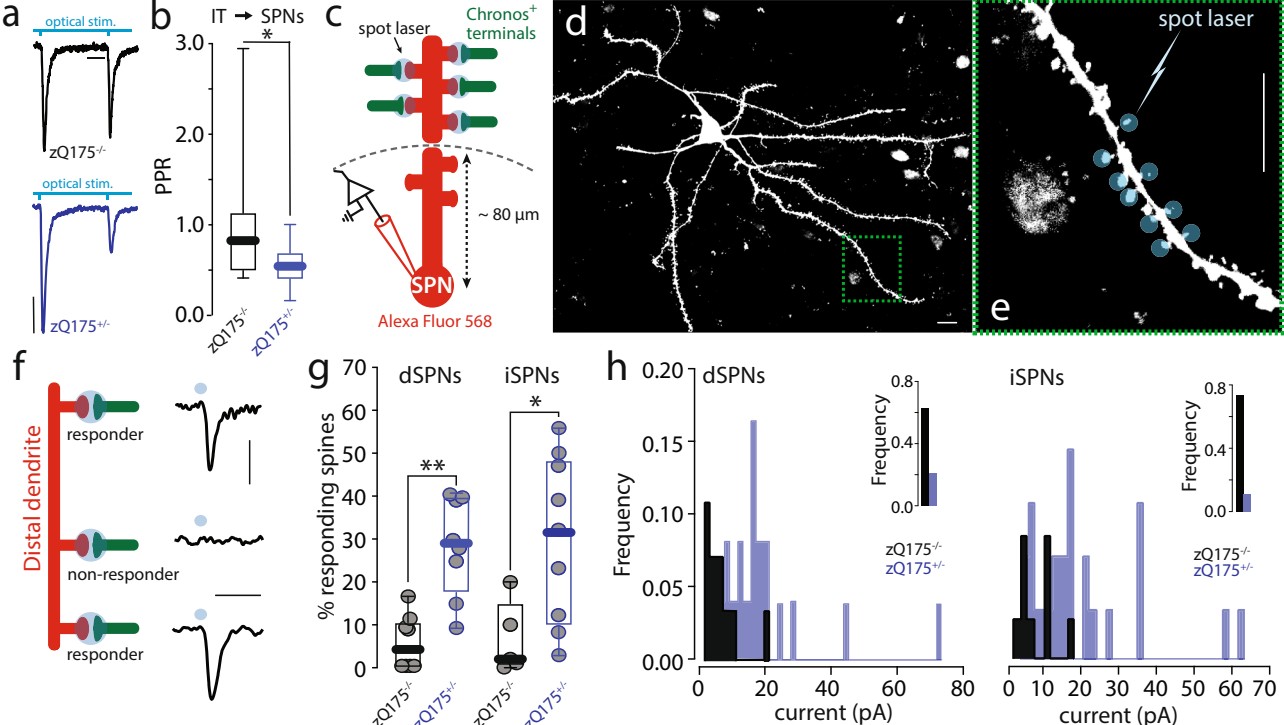

**Fig. 2 | Presynaptic changes in IT axo-spinous synapses in male zQ175+/-.**
**a** Sample optically evoked EPSCs (oEPSCs) evoked by a paired-pulse stimulation (PPR) of IT terminals (Scale bars: 25 ms and 50 pA; see Fig. 1a for AAV9.hSyn.-Chronos-GFP stereotaxic delivery). **b** Boxplot summary showing reduced PPRs (oEPSC$_2$/oEPSC$_1$) in zQ175+/- compared to zQ175-/- controls (zQ175-/- $N = 5$; $n = 16$ and zQ175+/- $N = 7$; $n = 26$; *$p = 0.011$ in a Mann–Whitney non-parametric two-sided test), boxplots represent median and interquartile range, whiskers min/max value. **c** Representation of a dendrite (proximal/distal sections shown) emanating from an SPN patched and dialyzed with Alexa Fluor 568. In green, contralateral IT terminals expressing chronos. Blue circles depict the spot laser beam (∅ ~ 1 μm, 475 nm, 1 ms duration) used for focal spine stimulation. **d** Representative dSPN filled with Alexa 568 (Scale bar: 10 μm; $N = 21$). **e** Parfocal distal dendrite. Each spine is being sequentially stimulated with one laser pulse (Scale bar: 10 μm; $N = 21$). **f** Traces of

somatic oEPSCs evoked by a single light pulse delivered on each visible parfocal spine. Figure shows spines generating a somatic oEPSC (responder) or not (non-responder spine; Scale bars: 20 ms and 20 pA). **g** Boxplots summarizing the results. Significant increase in the number of active axo-spinous synapses in the distal compartment of both dSPNs and iSPNs in zQ175+/- animals (boxplots represent median and interquartile range, whiskers min/max value; dSPNs zQ175-/- ($N = 7$; $n = 8$), zQ175+/- ($N = 6$; $n = 8$), **$p = 0.001$ in a Mann–Whitney non-parametric two-sided test; iSPNs zQ175-/- ($N = 4$; $n = 5$), zQ175+/- ($N = 4$; $n = 9$), *$p = 0.019$ in a Mann–Whitney non-parametric test. **h** histogram of oEPSC amplitudes in zQ175-/- compared to zQ175+/- in dSPNs (left) and iSPNs (right). Inserts represents the frequency of "no response" (0 pA). Frequency distribution is significantly different with a two-tailed Kolmogorov–Smirnov test in dSPNs and iSPNs ($p < 0.0001$).

were performed in contralateral dSPNs and presumed iSPNs (tdTomato-negative) as described above. In contrast to the experiments with PT axons, Oxo-M robustly reduced the peak amplitude of EPSC evoked by optical stimulation of IT axons. The modulation was observed in both dSPNs and iSPNs (Fig. 4e, f) and was accompanied by an increase in paired-pulse ratio (PPR) (Fig. 4g). Moreover, the modulation was persistent in both types of SPN (Fig. 4f).

To complement these pharmacological approaches, the ability of ACh release from ChIs to mimic the muscarinic modulation of IT terminals was determined using a combination of chemogenetic and optogenetic approaches. To chemogenetically activate ChIs, an AAV carrying a Cre-dependent, 5HT3 receptor-based 'pharmacologically selective actuator module' (PSAM-5HT3[45]) expression construct was injected into the DLS of mice expressing Cre recombinase under control of the choline acetyltransferase promoter (ChAT-cre) and tdTomato under control of the Drd1 promoter. In the same mice, an AAV carrying synapsin-driven Chronos-GFP expression construct was stereotaxically injected into the contralateral M1 cortex, as described above (Fig. 5a, b). A month later, ex vivo slices were prepared and whole-cell voltage clamp recordings obtained from dSPNs and presumed iSPNs as described above. PSAM-5HT3 activation was achieved by bath application of its cognate synthetic ligand (PSEM, pharmacologically selective effector module; 10 μM)[45]. In control recordings,

PSEM robustly increased ChI spiking (Fig. 5c, d). When recording from SPNs, PSEM induced a reduction in optically evoked IT oEPSCs (Fig. 5e, f). The modulation was indistinguishable in dSPNs and iSPNs (Fig. S14), therefore, the data from the two cell types was grouped. As with Oxo-M application, the modulation was long-lasting and persisted after washing out PSEM. As expected, this effect was also accompanied by an increase in PPR (Fig. 5g).

**Presynaptic modulation of IT terminals was mediated by M4 mAChRs**

To better understand the differential modulation of IT and PT axon terminals by ACh, molecular approaches were used. AAV9.CAG.DIO-tdTomato was stereotaxically injected into the M1 cortex of either Sim1-(KJ18)-Cre (PT-specific Cre recombinase expression), or Tlx3-(PL56)-Cre (IT-specific Cre recombinase expression) mice (Fig. 6a). A month later, M1 cortex was dissected out, dissociated, and then subjected to fluorescence-activated cell sorting (FACS). FACS isolated PT and IT neurons were then subjected to reverse transcription-quantitative polymerase chain reaction (RT-qPCR) analysis (Fig. S5a). This analysis revealed that M4 mAChR mRNA expression was roughly six-fold greater in IT than PT neurons (Fig. 6b). In contrast, the expression of M2 mAChR mRNA was relatively low and not different between the two cell types (Fig. 6b). To provide an independent

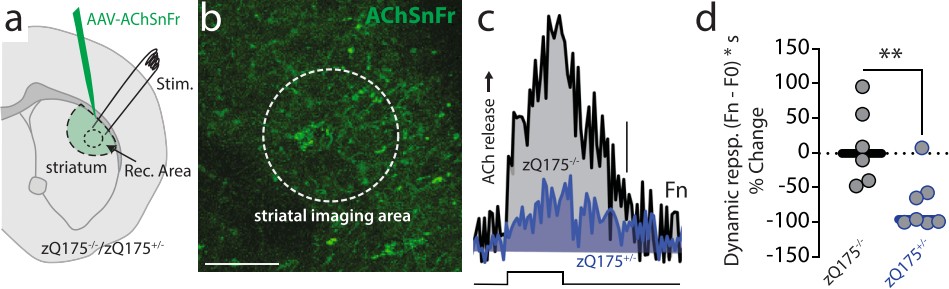

**Fig. 3 | Reduced evoked ACh release in DLS in male zQ175$^{+/-}$ mice. a** Diagram showing AAV-AChSnFr (Acetylcholine sniffer) stereotaxically injected in dorso-lateral striatum (DLS). The imaging area and bipolar electrical stimulator are indicated. **b** DLS section showing the striatal neuropil expressing AAV-AChSnFr and the imaging area (Scale bar: 50 μm). **c** Representative traces showing a decrease in evoked acetylcholine (ACh) released in zQ175$^{+/-}$ compared to zQ175$^{-/-}$ controls. Scale bar: 0.05 Normalized Fluorescence (Fn). Electrical stimulation (step) consisted of 20 − 1 ms stimuli, of 1 mA delivered at 2 Hz. **d** Boxplot summarizing the data from zQ175$^{-/-}$ ($N = 3$; $n = 6$ slices), zQ175$^{+/-}$ ($N = 4$; $n = 7$), **p = 0.0082 Mann−Whitney non-parametric two-sided test. Shown is the % change compared to median data collected in zQ175$^{-/-}$ controls, and the Dynamic response [(Fn − F0) * s], see "Methods".

confirmation of the functional significance of this difference, the M4 positive allosteric modulator VU0152100 (5 μM) was co-applied with a sub-saturating concentration of the non-specific mAChRs agonist Oxo-M (100 nM) while monitoring oEPSCs amplitude in SPNs evoked by optogenetic stimulation of IT axons. While it had no effect on its own (Figs. 6d and S5f), VU0152100 roughly doubled the amplitude of the presynaptic Oxo-M modulation (Fig. 6c, d) and enhanced the PPR compared to Oxo-M alone (Fig. S5g). Together, these data are consistent with the hypothesis that the modulation was mediated largely (if not exclusively) by M4Rs.

As mentioned above, one of the unexpected features of the IT modulation produced by cholinergic signaling was its persistence (Fig. 4f). Given that this effect was attributable to G$_{i/o}$-coupled M4Rs, there are two signaling pathways that these receptors might influence to control glutamate release. One is G$_{\beta\gamma}$-mediated inhibition of Cav2 Ca$^{2+}$ channels linked to exocytosis[46,47]. The change in PPR produced by M4R activation is consistent with inhibition of active zone Cav2 Ca$^{2+}$-channels[46]. But this membrane-delimited modulation typically rapidly reverses with agonist washout[48]. Another signaling linkage of M4Rs is G$_{\alpha i}$-mediated inhibition of adenylyl cyclase (AC) and protein kinase A (PKA) (Fig. S5e)[46,49]. Elevation of terminal cyclic adenosine monophosphate (cAMP) signaling is well known to enhance activity-dependent transmitter release[50]. To test for the involvement of cAMP signaling, the cell permeable cAMP analog 8-Br-cAMP was bath applied a few minutes before Oxo-M while monitoring IT oEPSCs. Indeed, 8-Br-cAMP stimulation of PKA significantly reduced the duration of the Oxo-M modulation, but not its initial amplitude (Figs. 6e, f, S5d). A similar result was obtained with bath application of the AC activator forskolin (Fig. S5b, d). Although postsynaptic M4Rs in dSPNs can lower glutamate release probability by engaging activity-dependent endocannabinoid signaling[51], pre-incubation with the CB1 selective antagonist PF514273 did not disrupt the ability of Oxo-M to attenuate IT synaptic transmission (Fig. S5c, d). Taken together, these data suggest that M4R activation on IT terminals rapidly and transiently suppresses glutamate release through a G$_{\beta\gamma}$ signaling pathway coupled to Cav2 Ca$^{2+}$ channels, but also persistently diminishes glutamate release through G$_{\alpha i}$-mediated inhibition of a constitutively active AC/PKA pathway (Fig. S5e)[46,52].

### Lowering mHtt in ChIs, but not cortex, normalized IT connectivity in zQ175$^{+/-}$ mice

The data presented until this point suggests that mHtt in zQ175$^{+/-}$ mice elevated glutamate release by IT synapses on SPNs either by attenuating M2/4R mediated presynaptic inhibition or by reducing ACh release by striatal ChIs. To determine which mechanism was responsible, an AAV vector was used to deliver an expression construct for a zinc finger protein (ZFP) targeting the expanded CAG repeat of mHtt[11].

To assess off-target effects of ZFP expression, the ZFP that binds to mHtt mRNA (bZFP) was compared to a non-binding ZFP variant (nbZFP)[11,31]. AAV9 vectors carrying synapsin promoter driven bZFP and nbZFP expression constructs were injected in the M1 cortex of 6−7-month-old Drd2-eGFP zQ175$^{+/-}$ mice (Fig. S6a, b). Two months later, RT-qPCR analysis of M1 cortex confirmed that the bZFP, but not the nbZFP[11], reduced mHtt mRNA abundance (Fig. S6c left); as previously described[11], wild-type Htt mRNA abundance was unaffected by either ZFP (Fig. S6d). In addition, there were no discernible alterations in the expression of mHtt or wild-type Htt contralateral to the bZFP injection site (Fig. S6c right). To assess the functional consequences of reducing cortical mHtt, the M1 cortex was injected with AAV8 vector carrying a Chronos expression construct one month after the AAV9 ZFP vector (Fig. S6a, b). A month after injecting the Chronos vector, ex vivo brain slices were prepared and the functional connectivity of IT neurons with SPNs assessed as described above. Although use of the AAV8 vector (to minimize the immunological response to a second virus injection) resulted in less robust Chronos expression and smaller evoked responses in SPNs, it was still clear that reducing mHtt in M1 cortex pyramidal neurons had no effect on IT evoked responses in SPNs (Fig. S6e−h). Next, the same ZFP vectors were injected into the DLS (Fig. S7a, b), and the functional connectivity of IT pyramidal neurons assessed (Fig. S7d−k). As with cortical targeting, the AAV reporter was broadly expressed by striatal neurons and the bZFP effectively lowered striatal (but not cortical) mHtt mRNA abundance (Fig. S7c)—leaving the wild-type Htt intact. In stark contrast to the effects of lowering mHtt in the cortex, lowering striatal mHtt normalized IT responses in SPNs (Fig. S7h−k).

The question left unresolved by these experiments was the striatal location of the mHtt responsible for the alteration in IT connectivity. Our working hypothesis was that ChI expression of mHtt was responsible. To test this hypothesis, two experiments were performed. First, mHtt was selectively lowered in ChIs of zQ175$^{+/-}$ mice and then ACh release compared to that in age-matched zQ175$^{-/-}$ littermate controls mice. To achieve cell-specific ZFP expression, ChAT-Cre x zQ175$^{+/-}$ mice (as above) were injected with AAV9 vectors carrying Cre-dependent ZFP or ZFP control expression constructs (AAV9.DIO-bZFP-eGFP and AAV9.DIO-nbZFP-eGFP, respectively). A month later, the AChSnFr AAV expression construct was delivered to the striatum (Fig. 7a, b) and a month later striatal ACh release was assessed as described above. Targeting the bZFP to ChIs normalized ACh release in zQ175$^{+/-}$ striata compared to data collected in zQ175$^{-/-}$ littermate controls, while the nbZFP control construct did not (Fig. 7d−g). To provide an added control, cortical ACh release was measured; it was similar in slices from zQ175$^{+/-}$ and zQ175$^{-/-}$ controls (Fig. S8), suggesting that the function of basal forebrain cholinergic neurons is not altered at this stage in zQ175$^{+/-}$ mice.

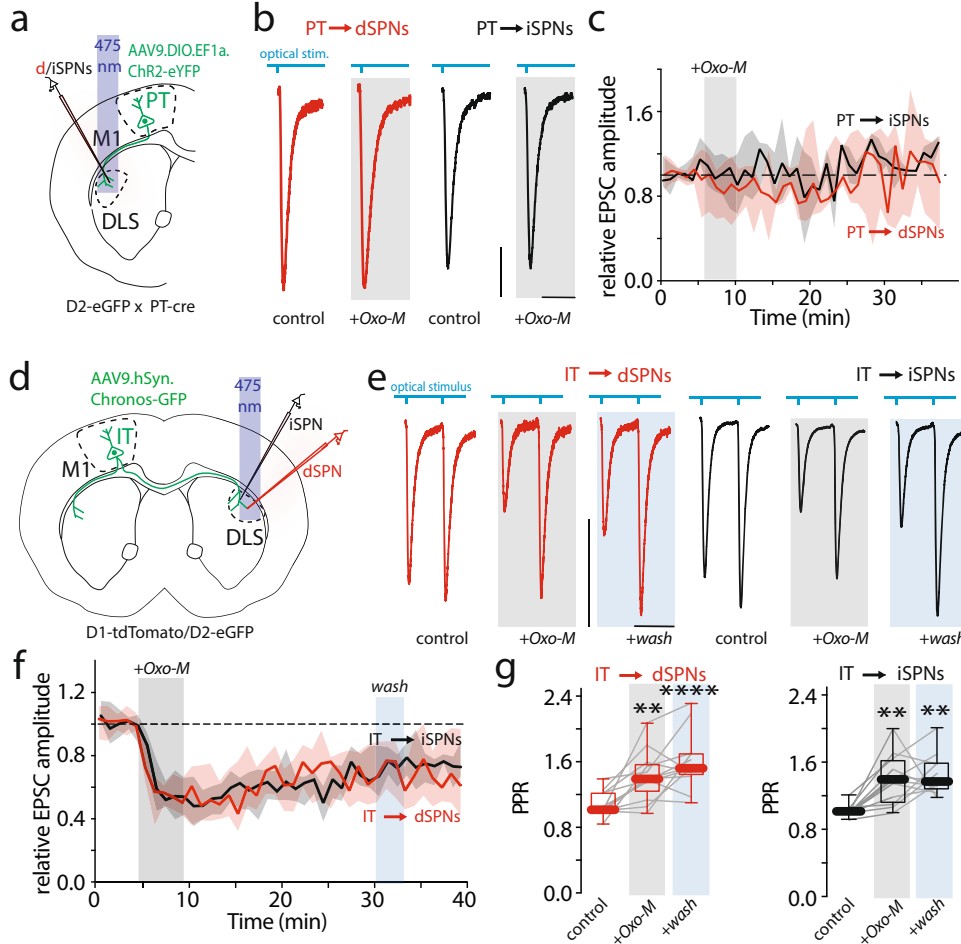

**Fig. 4 | Muscarinic activation modulates IT corticostriatal transmission, but not PT. a** Schematic of AAV9.EF1a.DIO.ChR2-GFP delivered in primary motor cortex (M1) of D2-eGFP crossed with PT-cre (Pyramidal tract-cre) male mice. Whole-cell recordings were obtained in DLS from ipsilateral SPNs. **b** Representative optically evoked EPSCs (oEPSC) traces from direct pathway SPNs (dSPNs, red) and indirect pathway SPNs (iSPNs, black) obtained by stimulating PT terminals. Traces acquired at baseline (control), and during Oxotremorine-M (Oxo-M, 10 μM) perfusion (Scale bars: 10 ms and 50 pA). **c** Time-series of oEPSC peak amplitudes normalized to control and plotted over time. No effect of five minutes application of Oxo-M (10 μM) on oEPSC amplitude. **d** Schematics of AAV9.hSyn.Chronos-GFP delivered in M1 of D1-tdTomato or D2-eGFP mice. Whole-cell recordings were obtained from SPNs in contralateral DLS. **e** Representative oEPSC traces obtained by stimulating

intratelencephalic (IT) terminals and recorded from dSPNs (red) and iSPNs (black) before, during, and after (*wash*) Oxo-M (10 μM) perfusion (Scale bars: 50 ms and 50 pA). **f** Five minutes Oxo-M application induces a similar and long-lasting reduction of oEPSC amplitude in both dSPNs and iSPNs. **g** Significant increase in paired pulse ratio (PPR) ratio during Oxo-M perfusion and during the *wash* period. In dSPNs, control vs. Oxo-M ($N = 4$; $n = 13$), **$p = 0.0650$; control vs. wash ($N = 4$; $n = 13$), ****$p < 0.0001$. In iSPNs, control vs. Oxo-M ($N = 4$; $n = 13$), **$p = 0.0019$; control vs. wash ($N = 3$; $n = 8$), **$p = 0.0090$ in a Kruskal-Wallis test with Dunn's multiple comparisons. Boxplots represent median and interquartile range, whiskers min/max value. In **c** and **f**, shading represent the interquartile range and line the median. All mice used were male.

Next, the functional connectivity of IT neurons with SPNs was assessed after selectively reducing mHtt in ChIs. If a reduction in cholinergic signaling was responsible for the corticostriatal pathophysiology in zQ175[+/−] mice, then normalizing ACh release by expressing bZFPs in ChIs should correct it. To test this hypothesis, the striatum of ChAT-Cre x zQ175[+/−] mice was injected with an AAV9 vector carrying Cre-dependent bZFP and nbZFP control expression constructs. An AAV8 vector was used to deliver a Chronos expression construct to the contralateral M1 cortex (Fig. 7h) as described above. The selectivity of the targeted reduction in mHtt in ChIs was verified using a validated RNAScope probe (Fig. S10)[53]. As predicted, lowering mHtt selectively in ChIs normalized IT responses in zQ175[+/−] SPNs (Fig. S8i–l). This effect was accompanied by an enhanced PPR in ZFP-injected striata compared to nbZFP controls (Fig. S11).

To provide an additional test of the cellular specificity of the effect on IT connectivity with SPNs, the bZFP construct was expressed in iSPNs and then IT connectivity assessed. To this end, Adora2a-Cre mice (a mouse line expressing Cre in iSPNs neurons) were crossed into the

zQ175[+/−] line. The striatum of Adora2a-Cre x zQ175[+/−] mice was then injected with the Cre-dependent AAV9 ZFP vector and two months later IT connectivity assessed with optogenetic tools as described above (Fig. S12a, b). Lowering mHtt in iSPNs did not significantly alter the enhanced response to optogenetic stimulation of contralateral M1 cortex IT axons in zQ175[+/−] (Fig. S12c, d).

## Discussion

There are four conclusions that can be drawn from the experimental results presented. First, in the zQ175[+/−] model of HD, the functional connectivity of IT−but not PT−cortical pyramidal neurons with both iSPNs and dSPNs increases as behavioral deficits begin to emerge[54]. Second, this upregulation is attributable to de-depression of IT synaptic terminals. Third, this de-depression is due to a deficit in ACh release by zQ175[+/−] ChIs, resulting in reduced inhibitory presynaptic M4R signaling. Lastly, selectively lowering mHtt in zQ175[+/−] ChIs with a virally-delivered ZFP expression construct restores ACh release and normalizes IT synaptic connectivity. Taken together, these studies

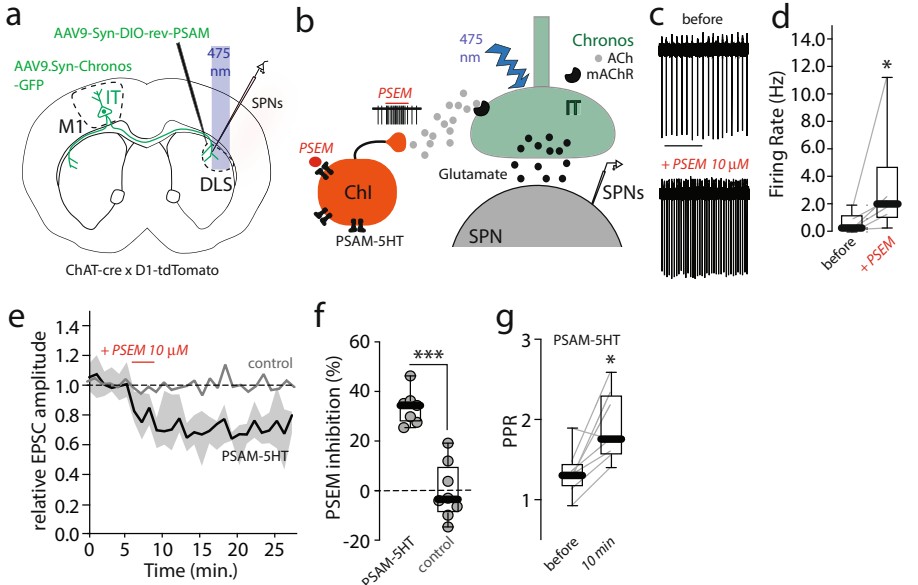

**Fig. 5 | Endogenous acetylcholine release inhibits IT corticostriatal transmission. a** We stereotaxically injected an AAV vector carrying 5HT receptor-based pharmacologically selective actuator module (PSAM-5HT)—associated to a Cre-dependent expression construct (AAV.DIO.PSAM.5HT3-GFP)—in dorsolateral striatum (DLS) of ChAT-cre mice. In the same mice, we injected an AAV vector carrying the AAV9.hSyn.Chronos-GFP construct in primary motor cortex (M1) contralateral to the PSAM-injected striatum. Thirty days post injection, we performed whole-cell recordings in SPNs from the striatum expressing PSAM-5HT. **b** Cartoon depicting pharmacologically selective effector module (PSEM) activation of PSAM-5HT-expressing cholinergic interneurons (ChI). Presynaptic activation of muscarinic acetylcholine receptors (mAChRs) reduces glutamate release from intratelencephalic (IT) terminals. **c, d** Strong increase in firing rate of ChIs expressing PSAM achieved by bath application of PSEM (10 μM). **c** Representative traces obtained in cell-attached configuration (Scale bar: 4 s). **d** boxplot summary of the results (*$p = 0.0312$, in a Wilcoxon two-tailed paired test; $N = 3$, $n = 6$ pairs). **e** Time-series data describing the prolonged optically evoked EPSCs (oEPSC) inhibition obtained with three minutes bath application of PSEM (10 μM) when recording from SPNs (control $N = 4$, $n = 9$; PSAM-5HT-injected mice $N = 4$, $n = 9$; shading represent the interquartile range and line the median). **f** Inhibitory effect of PSEM on oEPSCs amplitude in slices from mice injected with AAV.DIO.PSAM.5HT3-GFP compared to non-injected mice (control). The inhibitory effect of PSEM is present only in PSAM-5HT injected mice (at 20–23 min after PSEM application: PSAM-5HT $N = 3$, $n = 7$; control $N = 4$, $n = 8$; ***$p = 0.0003$ in a Mann–Whitney non-parametric two-tailed test). **g** PSEM stimulation of ChIs increases oEPSCs paired pulse ratio (PPR) recorded in SPNs compared to baseline control (before; *$p = 0.0156$, in a Wilcoxon paired two-tailed test; $n = 8$ pairs). All Boxplots represent median and interquartile range, whiskers min/max value. All mice used were male.

provide new insight into the network mechanisms underlying HD, as well as how cholinergic signaling regulates interhemispheric flow of information from the cerebral cortex to the striatum.

Striatal release of ACh by ChIs has long been known to suppress glutamate release by corticostriatal terminals synapsing upon SPNs[21,24,25]. However, the selectivity and persistence of the modulation has not been appreciated. Using optogenetic approaches to selectively stimulate ipsilateral or contralateral IT inputs to SPNs, our studies demonstrate that M2-class mAChR signaling (particularly M4) potently inhibits glutamate release[25]. This inhibition was accompanied by an elevation in paired-pulse ratio, a hallmark of a presynaptic mechanism. In contrast, mAChR activation had no effect on PT axon terminals studied using the Sim1-KJ18 transgenic line of mice. This differential modulation was attributable to the preferential expression of M4Rs in IT neurons, as RT-PCR profiling revealed robust M4R mRNA expression in IT, but not PT, neurons and an M4R selective PAM enhanced the modulation.

As for a number of other G$_{i/o}$-coupled GPCRs on corticostriatal terminals[46,55,56], activation of M4Rs produced a persistent suppression of glutamate release. The persistence was not an artifact of applying exogenous mAChR ligands, as transient chemogenetic activation of ChIs produced similar persistent suppression of synaptic transmission. Although the mechanisms responsible for the persistence of the modulation remain to be fully elucidated, the ability of targeted manipulation of AC signaling to control this slow phase of the modulation is consistent with the proposition that transient M4R activation can have a long-lasting effect on constitutive PKA mediated enhancement of activity-dependent exocytosis[46]. This feature of the M4R

modulation, and that of other G$_{i/o}$-coupled GPCRs, merits additional study.

In agreement with inferences drawn from previous studies, evoked ACh release from zQ175$^{+/-}$ ChIs measured using a recently developed optical probe (AChSnFr)[44] was significantly lower than that from control zQ175$^{-/-}$ ChIs. The same deficit in ACh release was seen in the R6/2 HD model, demonstrating that it wasn't peculiar to the zQ175 model. The deficit in ChI signaling provided a simple explanation for the enhanced functional connectivity of IT neurons with SPNs. Although our release experiments focused on the DLS, the enhancement in connectivity was evident at IT projections from M2 and CgCtx to the DMS as well. Even though the cholinergic innervation of the striatum is not limited to ChIs[57], their causal role in the alteration in IT connectivity was cemented by the observation that knocking down mHtt selectively in ChIs normalized the responsiveness of SPNs to IT stimulation in zQ175$^{+/-}$ striata. Moreover, knocking down mHtt generically in M1 cortical neurons or selectively in iSPNs did not affect IT synaptic function.

A presynaptic locus for the change in IT connectivity in zQ175$^{+/-}$ mice was supported by three lines of evidence. First, the ratio of currents through AMPA and NMDA receptors was indistinguishable at IT synapses on zQ175$^{+/-}$ and control SPNs, suggesting that postsynaptic potentiation was not a factor. Second, using Chronos to stimulate IT axons repetitively, it was found that the release probability at IT synaptic terminals was elevated in ex vivo brain slices from zQ75$^{+/-}$ mice, consistent with a loss of inhibitory presynaptic M4R signaling. Third, the number of dendritic spines having a detectable response to presynaptic optical stimulation of IT terminals was significantly greater

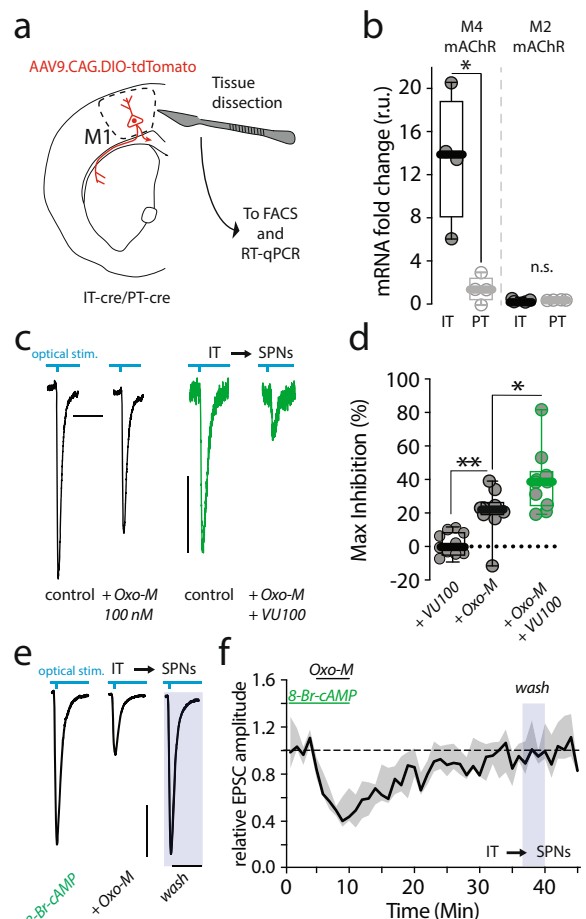

**Fig. 6 | M4 modulation of IT corticostriatal transmission. a** Schematic of AAV9.CAG-DIO-tdTomato stereotaxically delivered in primary motor cortex (M1) in Sim1(KJ18)-Cre (PT-cre) and Tlx3(PL56)-Cre (IT-cre) male mice. M1 tdTomato positive are enzymatically separated, sorted with Fluorescence-activated cell sorting (FACS), and analyzed (see also Fig. S4a and Materials and Methods). **b** Boxplot showing that M4 muscarinic receptor (mAChR) mRNA in IT neurons is significantly higher compared to PT (IT $N = 4$ animals, PT $N = 4$ animals; $*p = 0.013$ in a Mann–Whitney non-parametric test). No difference in M2 expression. **c** Representative optically evoked EPSCs (oEPSC) traces recorded from spiny projection neurons (SPNs) and obtained by stimulating intratelencephalic (IT) terminals as described in Fig. 1a (Scale bars: 50 ms and 100 pA). Traces acquired before (control), and during 5 min bath application of a sub-maximal concentration of Oxo-M (100 nM). In the experiments with the M4 positive allosteric modulator (PAM) VU0152100 (green, VU100, 5 µM), VU100 was pre-applied for at least 15–20 min before adding Oxotremorine-M (Oxo-M). **d** Boxplots summarizing data showing that bath application of VU100 potentiates the inhibitory effects of Oxo-M (100 nM) on IT transmission, while it has no effect by itself (**$p = 0.001$, *$p = 0.0185$ in Mann–Whitney non-parametric two-tailed test, $N = 3$; $n = 10$ for all groups). **e** Representative oEPSC traces recorded from SPNs and obtained by stimulating IT terminals (Scale bars: 20 ms and 50 pA). 15–20 min pre-incubation with the cell-permeable cAMP analog 8-Br-cAMP blocked the long-term depression (LTD) obtained with 5 min. application of Oxo-M (10 µM). **f** Time-series data describing the effects of pre-incubating the slices with 8-Br-cAMP on the chemically induced, muscarinic-dependent long-term depression (LTD, shading represent the interquartile range and line the median; see also Fig. S4). No significant effect of 8-Br-cAMP was seen on the acute inhibition. All mice used were male. All Boxplots represent median and interquartile range, whiskers min/max value.

in zQ175$^{+/-}$ SPNs. Given that previous studies have not found substantial alterations in SPN synapse density in zQ175 and BACHD models[6,40,58], this change was attributable to a shift in the amplitude distribution of postsynaptic responses, moving a significant number of

spines above the experimental detection threshold. Given that the postsynaptic complement of AMPA receptors had not changed and glutamate release at corticostriatal synapses is multi-vesicular and does not saturate postsynaptic glutamate receptors[21], this shift can be ascribed to removal of a persistent presynaptic inhibition of glutamate release. It is also worth noting that enhanced glutamate release at zQ175$^{+/-}$ IT terminals may provide an additional explanation for the reported engagement of extra-synaptic glutamate receptors in HD models[59].

How can our results be reconciled with previous reports of impaired corticostriatal synaptic transmission in HD models? Several methodological factors may be responsible. One of these factors is age of the model. The shift in IT connectivity was developmentally regulated, appearing only about the time zQ175$^{+/-}$ mice begin to manifest deficits[54]. Another factor was our use of optogenetic approaches to selectively activate IT axons, whereas previous studies have relied upon electrical stimulation that engages not only IT and PT corticostriatal axons, but also other glutamatergic axons (e.g., thalamostriatal), as well as neurons intrinsic to the striatum. This clearly complicates the interpretation of these previous results. For example, electrical stimulation might have led to preferential activation of PT axons, which are larger and have a lower threshold for electrical stimulation[18]; as glutamate release by PT terminals is enhanced by ACh release and nAChR activation[22] rather than being inhibited by mAChRs, their preferential recruitment in HD models—where ACh release is reduced—could have led to the appearance of impaired corticostriatal connectivity.

Another factor to consider is the extent to which postsynaptic mechanisms were engaged in assessing functional connectivity. In our experiments, neurons were loaded with Cs$^+$ and voltage-clamped to minimize the contribution of postsynaptic mechanisms, but this was not the case in previous studies of HD models[60]. Previous work by our group has shown that not only is postsynaptic LTP at glutamatergic synapses compromised in zQ175$^{+/-}$ iSPNs, but that dendritic excitability also is reduced[11,40]. The deficit in cholinergic signaling described here should work together with these cell-autonomous deficits induced by mHtt, as mAChR signaling enhances the dendritic excitability of iSPNs and promotes LTP induction[24,61]. Thus, it is reasonable to infer that, at least in iSPNs, spiking induced by electrical stimulation in ex vivo brain slices from HD models could be less robust, despite the dis-inhibition of IT terminals.

How does the recognition that mHtt impairs ChI ACh release and enhances IT connectivity with SPNs impact our understanding of HD? Several lines of study suggest that ChIs are important to set-shifting; that is, changing action selection with changing environmental circumstances or contingencies[24,62]. The mechanisms mediating set-shifting are far from being completely understood, but it does appear that it involves changing the balance in excitability of iSPNs and dSPNs toward iSPNs and cessation of on-going thoughts and actions[24]. The cell-autonomous effects of mHtt on the excitability of iSPNs and the ability to potentiate corticostriatal glutamatergic synaptic transmission should work together with the deficit in ChI signaling to create a 'perfect storm' in the indirect pathway, disrupting the ability to suppress unwanted actions. Unfortunately, testing this hypothesis in the zQ175 model is problematic, as, despite its faithful recapitulation of key HD features at the cellular and circuit level, it does not robustly reproduce the evolution of motor disability in HD patients[37,63].

What is less clear is how biasing the functional connectivity of SPNs toward IT and away from PT pathways fits into this scenario. It is thought that the PT pathway carries information about ongoing actions, whereas the IT pathway conveys information about internal state, sensory context, and motor planning[19,20]. Thus, transient ChI suppression of IT synapses could serve to diminish the influence of previously reinforced, contextual cues and planned actions on striatal ensembles, allowing new striatal ensembles to emerge that could drive

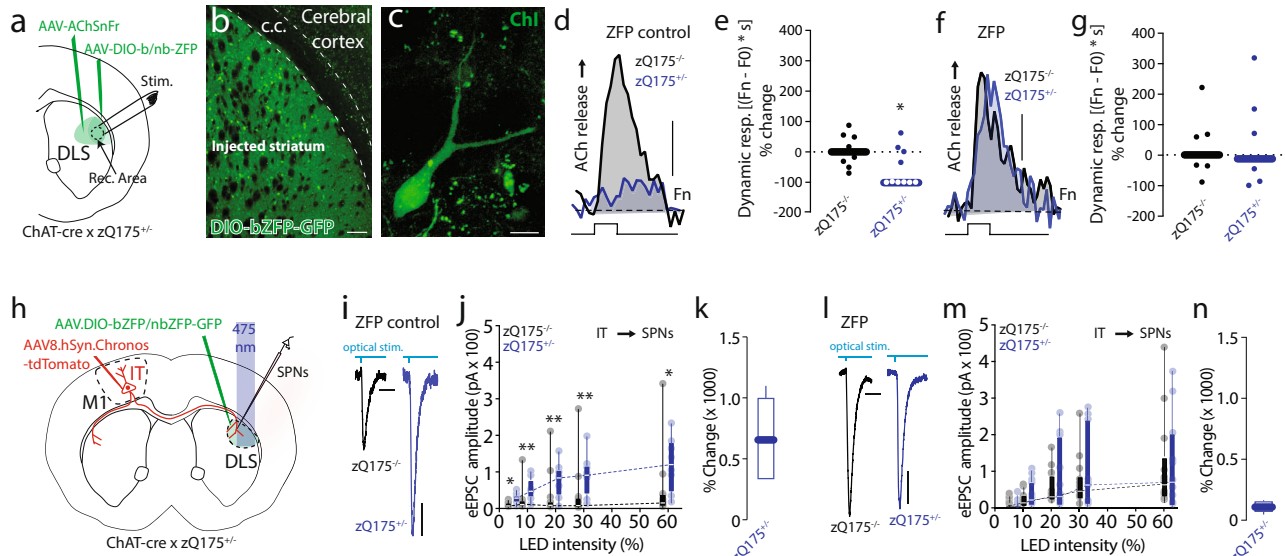

**Fig. 7 | Lowering mHtt in ChIs normalizes ACh release and IT connectivity in male zQ175+/− mice. a** Schematic of AAV.AChSnFr and AAV.DIO.ZFP/ZFP control-eGFP (ZFP, zinc finger protein) viral constructs injected in dorsolateral striatum (DLS) of ChAT-cre x zQ175+/− mice. **b** Confocal image of DLS (Scale bar: 100 μm) showing high expression of the AAV.DIO-bZFP-eGFP construct in ChAT-cre neurons **c** (Scale bar: 20 μm; N = 14). **d, e** Mice expressing ZFP control construct in cholinergic interneurons (ChIs) display reduced ACh release. Representative traces are shown in (**d**) (Scale bar: 0.1, Fn: normalized fluorescence), while (**e**) shows summary bar-graph of the dynamic response - zQ175−/− controls (N = 4, n = 8 slices), zQ175+/− (N = 4, n = 10), *p = 0.04343 in a Mann−Whitney non-parametric two-sided test. **f, g** ZFP expression in ChIs normalizes evoked ACh release. Representative traces in (**f**) (Scale bar: 0.1) while (**g**) shows summary data showing no difference in ACh release in - zQ175−/− (N = 3, n = 7 slices), compared to zQ175+/− (N = 3, n = 7 slices), p > 0.9999. All tests are Mann−Whitney non-parametric tests. **h** Schematic depicting the experimental paradigm used for virus delivery and Spiny projection

neurons (SPN) whole-cell recording in ChAT-cre x zQ175+/−. **i–k** Expression of the ZFP control construct selectively in ChAT+ neurons in DLS did not normalize IT corticostriatal transmission. Representative traces of oEPSCs elicited optically stimulating IT axons, **j** I/O curves obtained in zQ175−/− compared to zQ175+/− - zQ175−/− (N = 4, n = 14), zQ175+/− (N = 3, n = 8) p-value (Stim. Intensity, % of max): *0.0016 (5%), **0.0063 (10%), **0.0081 (30%), *0.0081 (60%) in a multiple Mann−Whitney non-parametric test. **l–n** Expression of the ZFP construct in ChAT-cre neurons normalized IT transmission in zQ175+/−. No difference is seen in I/O response in cells from zQ175−/− controls compared to zQ175+/−. zQ175−/− (N = 4, n = 13), zQ175+/− (n = 11) p-value (Stim. Intensity, % of max): 0.5309 (5%), 0.6084 (10%), 0.8646 (20%), 0.8646 (30%), 0.9546 (60%) in a multiple Mann−Whitney two-sided non-parametric test (Scale bar in **i** and **l**: 10 ms 50 pA). **k, n** % change compared to controls at 60% LED intensity. All Boxplots represent median and interquartile range, whiskers min/max value.

alternative, more adaptive actions. A presynaptic mechanism with a slow time course could be a particularly effective means of 'disrupting' striatal ensembles without completely erasing memories stored in postsynaptic circuitry. Regardless, the mechanisms for set-shifting and adaptive behavior should be impaired in HD. Indeed, a hallmark of HD is perseverative, obsessive behavior[64]. This inference is of considerable translational value as ZFP-mediated reduction in mHtt expression in symptomatic mice was able to normalize ACh release and the functional connectivity of IT neurons with SPNs. However, the fact that mHtt drives other, cell-autonomous forms of pathology, particularly in iSPNs, suggests that broadly targeting mHtt in the striatum to include SPN, ChIs, and astrocytes[11,65] with ZFPs[11,31], or other strategies like CRISPR-Cas9[66], is likely to be the most effective therapeutic path forward for HD patients.

## Methods
### Mice
All animal procedures were performed according to the Northwestern University Animal Studies committee and in accordance with the National Institutes of Health Guide for the Care and Use of Laboratory Animals. Mice were group housed with food and water provided ad libitum under a 12 h–12 h light–dark cycle and temperatures of 18–25 °C with 40–60% humidity. We used male mice heterozygous for the zQ175 knock-in (zQ175+/−) and littermate controls (zQ175−/−). zQ175+/− display an expanded polyglutamine sequence of ~190 repeats on a C57/Bl6 background B6J.zQ175DN KI mice (Jackson Labs, Stock No. 029928). To study the PT striatal afferents, we used Sim1-kj18-cre mice (MGI:4367070). For the FACS sorting experiments, we also used the Tlx3(PL56)-Cre line (MGI:5311700) to investigate IT neurons mRNA

expression (http://www.gensat.org/index.html). Mice were crossed with bacterial artificial chromosome (BAC) D1 dopamine receptor-tdTomato (D1-tdTomato; B6.Cg-Tg(Drd1a-tdTomato)6Calak/J; Jackson Labs, Stock No. 016204) or D2 dopamine receptor- eGFP (D2-eGFP; B6;FVB-Tg(Drd2-EGFP/Rpl10a)CP101Htz/J; 030255) lines as specified in the text and figures to identify dSPNs and iSPNs. Both lines have been backcrossed to pure C57/BL6 background. To manipulate cholinergic interneurons activity, we used ChAT-Cre mice (B6;129S60-Chat^tm2(cre) lowl/J; Jackson Labs, Stock No. 006410). These mice express Cre recombinase in cholinergic neurons, without disrupting endogenous Chat expression. To selectively target iSPNs we also used A2a-Cre mice, a mouse line expressing Cre selectively in iSPNs (Tg(Adora2a-Cre) KG139Gsat; MGI:4361654).

### Slice preparation
In accordance with Northwestern University Animal Studies committee, two to three and seven to nine months old mice were deeply anesthetized with ketamine and xylazine, intracardially perfused with ice-cold low-Ca²⁺ artificial cerebrospinal fluid (cutting ACSF) containing 125 mM NaCl, 7 mM glucose, 25 mM NaHCO₃, 2.5 mM KCl and 1.25 mM NaH₂PO₄, 0.5 mM CaCl₂, 2 mM MgCl₂ equilibrated with 95% oxygen and 5% CO₂. Coronal slices, 275 μm were cut in cutting ACSF, then incubated at 32 °C for at least 30 min in regular ACSF of similar composition to the cutting ACSF but with 2 mM CaCl₂ and 1 mM MgCl₂. Slices were subsequently stored at R.T. (-25 °C).

### Stereotaxic injections
Stereotaxic-guided surgeries were performed on mice anesthetized with isoflurane. After positioning the head to obtain a flat skull

between bregma and lambda, a small hole was drilled with a micro drill, and a glass pipette was slowly inserted at the following coordinates. Primary motor cortex (M1) injection coordinates were reached using a computer-guided stereotaxic instrument with atlas integration (Angle Two system, Leica) and were (relative to bregma): 1.2 mm AP, 1.6 mm ML, and −1.4 mm DV. For cingulate cortex (Cng. Ctx.): 1.34 mm AP, −0.3 mm ML, and −1.8 mm DV. For secondary motor cortex (M2): 1.94 mm AP, −0.75 mm ML, and −1.35 mm DV. For dorsolateral striatum (DLS) coordinates: 1.00 mm AP, 1.83 mm ML, and −3.04 mm DV.

### Electrophysiology
Slices were perfused at 2–3 ml/min with oxygenated regular ACSF containing 2 mM $CaCl_2$ and 1 mM $MgCl_2$ at -25 °C. For all experiments, 20 µM Gabazine and 2 µM CGP55845 were added to the perfusion media to block $GABA_A$ and $GABA_B$ receptors, respectively. SPNs restricted to the dorsolateral striatum were identified using infrared differential interference contrast on an upright Olympus Bx51WI (40X/0.8 NA objective) microscope and a CCD camera connected to a monitor. Somatic whole-cell recordings were made with borosilicate patch pipettes having open tip resistances of 3–5 MΩ. The intracellular pipette solution contained cesium-based internal solution to reduce the impact of synapses position and maximize voltage control: 115 mM cesium methylsulfonate, 5 mM HEPES, 5 mM tetraethyammonium-Cl, 2 mM QX-314, 0.25 mM EGTA, 2 mM Mg-ATP, 0.5 mM Na-GTP, 10 mM sodium phosphocreatine, pH 7.3, and osmolarity 280–290 mOsm. After rupture of the membrane the cells were allowed to dialyze with the internal solution for -15 min. Recordings were obtained using a MultiClamp 700A amplifier and pClamp 10 software. Pipette offset and capacitances were adjusted using MultiClamp 700B Commander software. Data were acquired at 10 kHz, filtered at 1 kHz using an 8-pole Bessel filter, and digitized using a DigiData 1322A 16-bit A/D converter. Somatic access resistance was monitored "offline" and "online" by delivering a 200 ms, −20 mV step in each sweep. Cells were held at −70 mV, not corrected for liquid junction potential. Input/output data from Cingulate cortex (Cng. Ctx.) and M2 (Fig. S1j–q) were obtained using a different setup. The primary difference was object lens, 60X/1.0NA, and the stimulation and display of the electrophysiological recordings that were obtained with freeware WinFluor software (John Dempster, Strathclyde University, Glasgow, UK).

Patch pipettes (3–6 MΩ) were prepared with a Sutter Instruments horizontal puller using borosilicate glass with filament and filled with the internal solution mentioned above. Access resistances were continuously monitored, and experiments were discarded if changes >20% were observed. Digitized data were imported for analysis with commercial software (IGOR Pro 6.0, WaveMetrics, Oregon).

### Optogenetics
To activate channelrhodopsin (ChR2) or Chronos containing axons in the dorsolateral striatum, we generated 475 nm/30 mm bandwidth light pulse with a pE 100 LED system (Cool Led) or an X-Cite 110LED Illumination System reflected through an eGFP filter cube (see Supplementary Table 1 for stimulation power). The stimulation light passed through a 40x objective lens (or 60x, for the Cng. Ctx. and M2) to produce a 500 µm (or 367 µm) diameter column of light at the slice surface. The microscope epi-illumination condenser iris on each system was reduced to limit the maximum sample power. The LED power setting was changed manually before stimulation and carefully titrated for physiological response. Unless otherwise specified, blue light pulses (3 ms duration) were triggered through pClamp via TTL. For most of our experiments and to obtain reliable PPR data, we used Chronos rather than ChR2, since ChR2 channels display slow recovery kinetics therefore it was impossible to reach stimulation frequencies necessary for PPR studies without generating a consistent depression of the second EPSC. One limitation of our approach could originate from the "over-bouton" stimulation paradigm used here. This choice was

made to reduce variability due to the relatively regional expression of the opsin in a distant of the contralateral cortex (M1, M2, or Cng. Ctx), and from the geometry of the axonal projections.

### Imaging
Large-scale montage brain slice images were acquired with the Olympus FV-10i-DUC automated confocal system with 10X/0.4NA (default 2.5 Airy unit pinhole setting) objective lens and higher resolution images with the 60X/1.35NA (default 2 Airy unit pinhole setting) oil-immersion objective. Montage images were acquired with zoom 1 (10x = 1.278 mm FOV) and 1024 × 1024 pixels. The 473 nm and 559 nm lasers were used to provide the eGFP and tdTomato images. Dendritic morphology and spine location identification as well as dynamic imaging was performed with Prairie Technologies Ultima series 2-photon excitation microscope system. The system foundation was an Olympus Bx51WI microscope equipped with a 60x/1.0NA(2 mm WD) water-dipping objective lens. Stage, lens focus, and manipulator remote motion control provided by Luigs and Neumann FM-380 plus SM-7 controllers. 2-photon excitation laser was Chameleon Ultra1 (690 nm to 1040 nm) and sCRACM stimulation via Coherent OBIS FP473-LX laser (50 mW post-fiber). Dual channel green and red non-de-scanned detection provided by green (490 nm to 560 nm) GaAsP cathode PMT (H10770B, Hamamatsu) and red (580 nm to 625 nm) by multi-alkali cathode PMT (R3986, Hamamatsu). The frame scan parameters were typically set at 256 × 256 pixel/image with a frame period of -0.500 s.

### Subcellular channelrhodopsin-2-assisted circuit mapping (sCRACM)
After whole-cell configuration was reached, cell was dialyzed for at least 15–20 min with an internal solution containing Alexa Fluor 568 (25 µM). sCRACM experiments were performed in the presence of TTX 1 µM, 4-AP 10 µM to block propagated activity, along with $GABA_{A/B}$ antagonists. Fluorescence images were acquired with a 2-photon laser scanning microscopy (2PLSM) setup as described above with 780 nm laser wavelength and red emission channel. Dendritic branches more than -80 µm from the soma ("distal") were selected only after ensuring that no other fluorescent dendrites of the patched cell were present above or below the branch tested. In each round of stimulations, each parfocal spine was stimulated with 1-ms-long laser pulse (i.s.i. 2 s, Coherent Obis Laser 473 nm) delivering -0.36 mW after 60x/1.0NA objective lens. Parfocal dendritic branches where more than 70% of the spines responded were excluded to reduce false positives. The results of three stimulation rounds (the sequence interval was approximately 1 min) were averaged and included in the study. The interpretation of the proximal dendrites data was more problematic due to constraints relative to the sCRACM technique and the radial shape of the SPNs dendritic branches.

### Fluorescent imaging of AChSnFr
Acetylcholine release was evoked in acute slices by local electrical stimulation in DLS using a bipolar electrode (Micro Probes WE3ST32.0B10) and a Digitimer DS3 stimulator. Stimulator control was 1 mA, 1 ms x 20 pulses at 2 Hz (9.5 s long). The external ACSF was supplemented with Gabazine 10 µM and CGP54626 10 µM to eliminate GABAergic activity. Fluorescence images were acquired with a 2PLSM setup as described above with 950 nm laser wavelength and green detection channel. The probe's fluorescence values were obtained by averaging the gray values in a circular region of interest (ROI, about 100 µm in diameter, see Fig. 3b) in each frame, using the Prairie brightness over time (BOT) function or Image J (NIH), and a time-lapse of the fluorescence changes was built for each experiment. Each trace was normalized to the minimum fluorescence ($F_{min}$) obtained by perfusing $Cd^{2+}$ 2 mM to block vesicular release, and a maximum ($F_{max}$) obtained perfusing ACh 100 µM at the end of each experiment to obtain a Normalized Fluorescence trace. The formula used was:

Normalized Fluorescence ($F_n$) = ($F - F_{min}$)/($F_{max} - F_{min}$). $F_n$ traces were used to calculate the dynamic response of the acetylcholine transient [($F_n - F_0$) * time (s)]. Each dynamic response data point was calculated from the median of at least two-three ROIs/slice. $Cd^{2+}$ did not have a significant effect on baseline fluorescence, indicating that tonic ACh release is low in the ex vivo slices. The experiments in Supplementary Fig. 8 were not normalized traces, but instead the simple $\Delta f/f_0$ where $f_0$ value did have no-laser background subtraction applied.

### Quantitative real-time PCR analysis of M2/M4 mRNA expression

Experimental procedures and data processing for fluorescence-activated cell sorting (FACS) were as previously described with minor modifications[67]. We used PT-cre or IT-cre mice injected with an AAV-DIO-tdTomato or a AAV-DIO-eGFP virus. Thirty days post injection primary cortex tissue was isolated from 300 μm thick coronal slices and single-cell suspensions were generated using a combination of enzymatic and mechanical dissociation procedures (Fig. S5a). PT or IT neurons were separated on a cell sorter (BD FACSAria SORP system and BD FACSymphony S6 SORP system, purchased through the support of NIH 1S10OD011996-01 and 1S10OD026814-01), based on tdTomato fluorescence. Approximately 3000–7000 cells from cortical tissue collected from 2 mice/sample. Total RNA was isolated from each PT and IT sample using the RNA-easy Micro RNA extraction kit (QIAGEN) according to the manufacturer instructions. Due to the relatively low cell count in the samples, cRNA was amplified using Ambion WT expression kit (Life Technologies). Five hundred or less ng of total RNA was copied to cDNA using the QuantiTect Reverse Transcription Kit (Qiagen) in a final volume of 20 μl. Real-time quantitative PCR reactions were performed in a 7500 Fast Real Time PCR System (Life Technologies). PCR reactions were performed in duplicates in a total volume of 20 μl containing 1–2.5 μl of cDNA solution and 1 μl of Taqman probe of the specific gene (Life Technologies). GAPDH was also estimated in each sample to normalize the amount of total RNA (or cRNA) used to perform relative quantifications. Fold change compared to GAPDH was expressed in relative units (r.u.). The assay identifications used were: Chrm2; muscarinic receptor 2: Mm01701855_s1; Chrm4; muscarinic receptor4: Mm00432514_s1.

### Quantitative real-time PCR analysis of Htt mRNA expression

RNA was isolated using RNAeasy kit (Qiagen) from striatal and cortical tissue of mice injected with ZFP AAV. The RNA was reverse transcribed with Superscript IV VILO Master Mix (Thermo Fisher Scientific). Quantitative real-time PCR was performed using an ABI StepOnePlus Real Time PCR system with TaqMan Fast Edvanced Master Mix (Thermo Fisher Scientific). The relative abundance of different transcripts was assessed by quantitative PCR in triplicate. The following primers/TaqMan probes (Thermo Fisher Scientific) were used for PCR amplification: Hprt: Mm03024075_m1; wild-type-mouse Htt_fw: CAG GTCCGGCAGAGGAAC, Mut-mouse-Htt_Q175_fw: GCCCGGCTGTGG CTGA, Mut and wild-type Htt_rv*: TTCACACGGTCTTTCTTGGTGG (*wild-type and mutant Htt share the same reverse primer sequence), probe: TGCACCGACCAAAGAAGGAACTCTCA. Experimental Ct values were normalized to Hprt values using the formula: $\Delta Ct = CtHtt - Ct$(Hprt). The final expression levels were shown as $\Delta Ct$ values.

### Vectors and drugs

AAV9.Syn.Chronos-GFP.WPRE.bGH (Addgene 59170, $2.23 \times 10^{13}$ gc/ml), AAV9.EF1a.DIO.hChR2 (H134R)-eYFP.WPRE.hGH (Addgene 20298, $2.11 \times 10^{13}$ gc/ml) and AAV9.CAG.Flex.tdTomato.WPRE.bGH (Addgene 28306, $2.51 \times 10^{13}$ gc/ml), AAV9-Syn-DIO-rev-PSAM-L141F-Y115F-5HT 3HC-IRES-GFP-WPRE (PSAM-5HT, $2.04 \times 10^{13}$ gc/ml). were supplied by University of Pennsylvania Vector Core and used for contralateral IT recordings, ipsilateral PT recordings. Sangamo Bioscience zinc finger repressor protein plasmids ZFP-30645 (mHtt repressing ZFP; bZFP), and DNA binding deleted control plasmid ZFP-DBD (nbZFP) were

obtained via CHDI Foundation, recloned in pFastbac (pFB) AAV shuttle vectors to produce hSyn-NLS-ZFP-30645-2A-eGFP-WPRE-bGHpA, hSyn-NLS-ZFP-deltaDBD-2A-eGFP-WPRE-bGHpA, CMV-NLS-ZFP-30645-2A-td Tomato-Sv40pA, CMV-NLS-ZFP-deltaDBD-2A-tdTomato-SV40pA, and the Cre-dependent variants hSyn-DIO-NLS-ZFP-30645-2A-EGFP-WPRE-bGHpA and hSyn-DIO-NLS-ZFP-deltaDBD-2A-EGFP-WPRE-bGHpA, and packaged into AAV9 by Virovek": AAV9.hSyn.ZFP30645.Flag.EGFP. WPRE (bZFP; $2.11 \times 10^{13}$ vg/ml), AAV9.hSyn.ZFP.control.Flag.2 A.GFP.WPRE (nbZFP; $2.06 \times 10^{13}$ vg/ml), AAV9.CMV.ZFP30645.- Flag.TdTomato.WPRE (bZFP; $2.28 \times 10^{13}$ vg/ml), AAV9.CMV.ZFP.con-trol.Flag.TdTomato.WPRE (nbZFP; $2.03 \times 10^{13}$ vg/ml), and the cre-dependent variants: AAV9.hSyn.DIO.ZFP.control.2 A.TdTomato (DIO. nbZFP; $2.12 \times 10^{13}$ vg/ml), AAV9.hSyn.DIO.ZFP.30645.2A.TdTomato (DIO.bZFP; $2.13 \times 10^{13}$ vg/ml). All drugs were obtained from Hello Bio, Sigma, or Tocris. PSEM[89s] was obtained from S.M. Sternson (Howard Hughes Medical Institute) and Apex Scientific. Acetylcholine sniffers AAVs (AChSnFr; pAAV1/2-hSynap.iAChSnFR.X513 V9; $0.38 \times 10^{13}$ gc/ml) were kindly donated by Loren Looger at Janelia.

### RNAscope assay

Adeno-associated viruses (AAVs) carrying ZFP transcription factors (AAV9.DIO.ZFP-tdTomato-WPRE) or control constructs (AAV9.-DIO.nbZFP-tdTomato-WPRE) were injected into the right and left dorsal lateral striatum (respectively) of five ChAT-cre x zQ175[+/−] mice. Mice were sacrificed 7–8 weeks after injection. Brain tissues were then quickly collected, embedded in OCT compound, and flash frozen. Coronal brain sections (12 micrometers) were prepared using a cryo-stat, and slides with sections were stored at −80 °C till use. RNAscope in situ hybridization assays for fresh frozen sections were carried out according to ACDbio manufacturer´s manual for RNAscope Multiplex Fluorescent Detection Kit v2 (Catalog# 323110). The following RNA-scope probes were used in this study: Mm-Htt-intron1-O2 (Catalog# 575581) for mutant Huntingtin; Mm-Chat-C3 (Catalog# 408731-C3) for choline acetyltransferase; WPRE-O4-C2 (Catalog# 540341-C2) as sur-rogate for ZFP or nbZFP expression. Slides were then mounted in ProLong Diamond Antifade Mountant with DAPI (Invitrogen) and imaged with Olympus FluoView FV10i confocal laser scanning micro-scope system. To measure mHtt expression in DLS ChAT neurons that also expressed ZFPs, ChAT neurons were outlined to create ROIs, and mHtt mRNA signal particles inside the ROIs were counted. Background signals as measured in WT mice were subtracted from the nbZFP and ZFP groups. Results were normalized to the nbZFP group. To measure mHtt expression in non-ChAT cells in the DLS, rectangular areas void of ChAT neurons were selected and mHtt mRNA signal particles were counted. 24 areas and 22 areas from 5 mice were mea-sured for nbZFP and ZFP groups, respectively. Non-parametric Mann–Whitney tests were performed to determine the statistical significance.

### Data analysis

pClamp 10.0 data files were analyzed using Clampfit 10.0 (Molecular Devices), Igor Pro 6.0 (WaveMetrics) and Microsoft Excel (v16.68, Microsoft), FlowJo v10.8.1. ACh release data were also analyzed with Clampfit 10.0, or Microsoft Excel v16.68. Graphs and figures were assembled with GraphPad Prism 9.0 (GraphPad software) and Illus-trator (Adobe) 2022. Statistical analysis was performed in GraphPad Prism 9.0, using specific tests as described in each figure.

### Reporting summary

Further information on research design is available in the Nature Portfolio Reporting Summary linked to this article.

## Data availability

The data generated in this study are provided in the Supplementary Information/Source data file. All source data needed to evaluate the

conclusions in the paper are provided in the Source data file. Source data are provided with this paper.

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

## Acknowledgements

We thank S. Ulrich, Danielle Rae Showalter, Marisha Alicea and Christine Kamide for expert technical assistance and Dr. Loren L. Looger for the ACh.SnFr probe. This work was supported by the CHDI and JPB Foundations.

## Author contributions

This project was initiated by D.J.S., T.P., and V.B. T.P. performed all the electrophysiology and 2-photon microscopy experiments (except for fig. S1 m and q performed by MD). T.P. also analyzed, gathered, and discussed all the data with D.J.S. and V.B. P.G.R., J.K., and T.T. performed the PCR experiments. Z.X. performed the RNAscope test. Y.C. performed some of the virus stereotaxic injections. D.L.W. was responsible for the 2-photon imaging setup and experiments. The manuscript was prepared and written by T.P. and D.J.S. and approved by all coauthors.

## Competing interests

The authors declare no competing interests.
