## [Peer Review File · Nature Communications]

Cholinergic deficits selectively boost cortical intratelencephalic control of striatum in male Huntington's disease model miceEditorial Note: This manuscript has been previously reviewed at another journal that is not operating a transparent peer review scheme. This document only contains reviewer comments and rebuttal letters for versions considered at *Nature Communications*.

REVIEWERS' COMMENTS

Reviewer #1 (Remarks to the Author):

The authors have done an excellent job addressing the issues raised. I have no further concerns.

Reviewer #2 (Remarks to the Author):

The authors have done a great job of responding to the comments in the initial review. There are no recommendations for further revision.

It is worth noting for the authors' information that there is a non-ACH binding mutant construct for iAChSnFR, as shown in supplementary figure S6 of the online preprint.

Also, the argument against using single electrical stimuli to evoke robust ACh release is not convincing given the previous findings from the Dani and Cragg labs, and many others, that single electrical stimuli evoke dopamine release that is largely dependent on beta2-containing nicotinic ACh receptors.

Reviewer #3 (Remarks to the Author):

In the revised manuscript, Pancani et al. have adequately addressed the majority of my concerns. The authors were very responsive in adding new data and bringing more evidence for cell-specificity effect. The new results are convincing and have corroborated the findings of this interesting study.

REVIEWERS' COMMENTS

Reviewer #1 (Remarks to the Author):

The authors have done an excellent job addressing the issues raised. I have no further concerns.

Reviewer #2 (Remarks to the Author): The authors have done a great job of responding to the comments in the initial review. There are no recommendations for further revision. It is worth noting for the authors' information that there is a non-ACh binding mutant construct for iAChSnFR, as shown in supplementary figure S6 of the online preprint. Also, the argument against using single electrical stimuli to evoke robust ACh release is not convincing given the previous findings from the Dani and Cragg labs, and many others, that single electrical stimuli evoke dopamine release that is largely dependent on beta2-containing nicotinic ACh receptors.

Response: *Thanks for pointing out the availability of the non-binding iAChSnFR mutant. Insofar as the argument about using a single electrical stimulus to evoke ACh release is concerned; we have edited this section of the manuscript. As we are sure the reviewer is aware, there are several factors that contribute to the ACh signal (e.g., electrode placement and design, stimulus intensity, probe expression level, etc.). Our goal was to have a stimulus that 1) mimicked the naturally occurring low-frequency spiking of ChIs, and 2) gave us a reliable response across subjects which included HD mice, where ACh release was impaired.*

Reviewer #3 (Remarks to the Author):

In the revised manuscript, Pancani et al. have adequately addressed the majority of my concerns. The authors were very responsive in adding new data and bringing more evidence for cell-specificity effect. The new results are convincing and have corroborated the findings of this interesting study.